# Recent Updates of the M4Raw Dataset and Applications in Evaluating MRI Denoising Methods

**Yi Li**[*1]                                              2210417009@stumail.sztu.edu.cn
**Mengye Lyu**[*1]                                                      lvmengye@sztu.edu.cn
**Guanxiong Luo**[2]                            guanxiong.luo@med.uni-goettingen.de
**Jingwei Guan**[1]                                              guanjingwei@sztu.edu.cn
**Jingyu Li**[†1]                                                    lijingyu@sztu.edu.cn

[1] *Shenzhen Technology University, Shenzhen, China 1*

[2] *University Medical Center Göttingen, Göttingen, Germany 2*

**Editors:** Accepted for publication at MIDL 2024

## Abstract

This paper presents the new multi-channel k-space dataset M4Raw acquired using low-field MRI. The M4Raw dataset comprises brain data from 183 subjects, each with 18 axial slices and three contrasts: T1-weighted (T1w), T2-weighted (T2w), and fluid attenuated inversion recovery (FLAIR). Additionally, the paper provides a description of the recently released test subset, as well as various denoising methods applied to the M4Raw dataset, demonstrating its potential applications in image denoising. Multiple deep learning methods trained on the M4Raw dataset, including traditional denoising networks and those using transformer modules, have been employed, achieving high-quality denoising of low-field MRI images. The M4Raw dataset not only facilitates the development of data-driven methods for low-field MRI denoising but also serves as a benchmark dataset for comparing different methods.

**Keywords:** Low-field MRI, Image denoising, Data-driven methods.

## 1. Introduction

Training data are critical for the development of data-driven methods. In comparison to natural images, MRI datasets with complex/k-space data are scarce. To address this gap, we have released a new multi-channel k-space dataset, named M4Raw (Lyu et al., 2023), acquired using low-field MRI. Since the public release of the M4Raw dataset, several updates have been made, primarily involving the release of the Gradient Recalled Echo (GRE) data and the test subset. In this paper, we provide a description of the test subset and demonstrate its potential applications in the field of image denoising.

## 2. Methods

### 2.1. Data

The dataset M4Raw is a new multi-channel k-space dataset acquired using low-field MRI. It contains brain data from 183 subjects, each with 18 axial slices and 3 contrasts: T1-weighted

---

[*] Contributed equally

[†] Corresponding author

(T1w), T2-weighted (T2w), and fluid attenuated inversion recovery (FLAIR). Importantly, each contrast includes two or three repetitions, resulting in more than 1,000 volumes in total (Lyu et al., 2023).

Recently, we have released the GRE data and test subset. The test subset includes three common sequences: T1w and T2w data, each acquired with six repetitions, and FLAIR data, acquired with four repetitions. Each sequence consists of 18 slices, with a slice thickness of 5 mm and an in-plane resolution of $0.94 \times 1.23$ $mm^2$. As shown in Fig 1, compared to three repetitions, test subset with more repetitions will have higher Signal to Noise Ratio (SNR) and better image quality, which makes it a more suitable reference for assessing the effectiveness of denoising methods. Various denoising methods trained on the M4Raw training set will be validated on test subset.

## T1w in the test subset

Figure 1: Illustration of the impact of repetitions on image SNR.

### 2.2. Denoising method

Multiple denoising methods were trained on the M4Raw training subset and evaluated on the test subset. These methods include UNET (Ronneberger et al., 2015) , RESTORMER (Zamir et al., 2022), NAFNET (Chen et al., 2022), MPRNET (Zamir et al., 2021) and SCUNET (Zhang et al., 2023). During training, the single repetition image from 128 subjects was used as input, and the multi-repetition averaged image was used as the labels. Note that all the models were separately trained and evaluated for each contrast, which differs from our journal version where models were trained with all contrasts mixed.

## 3. Results and Discussion

Figure 2 presents the denoising results. Notably, the robustness of the SCUNET method is better than that of other data-driven and traditional methods, always obtaining the highest two scores for all types of data.

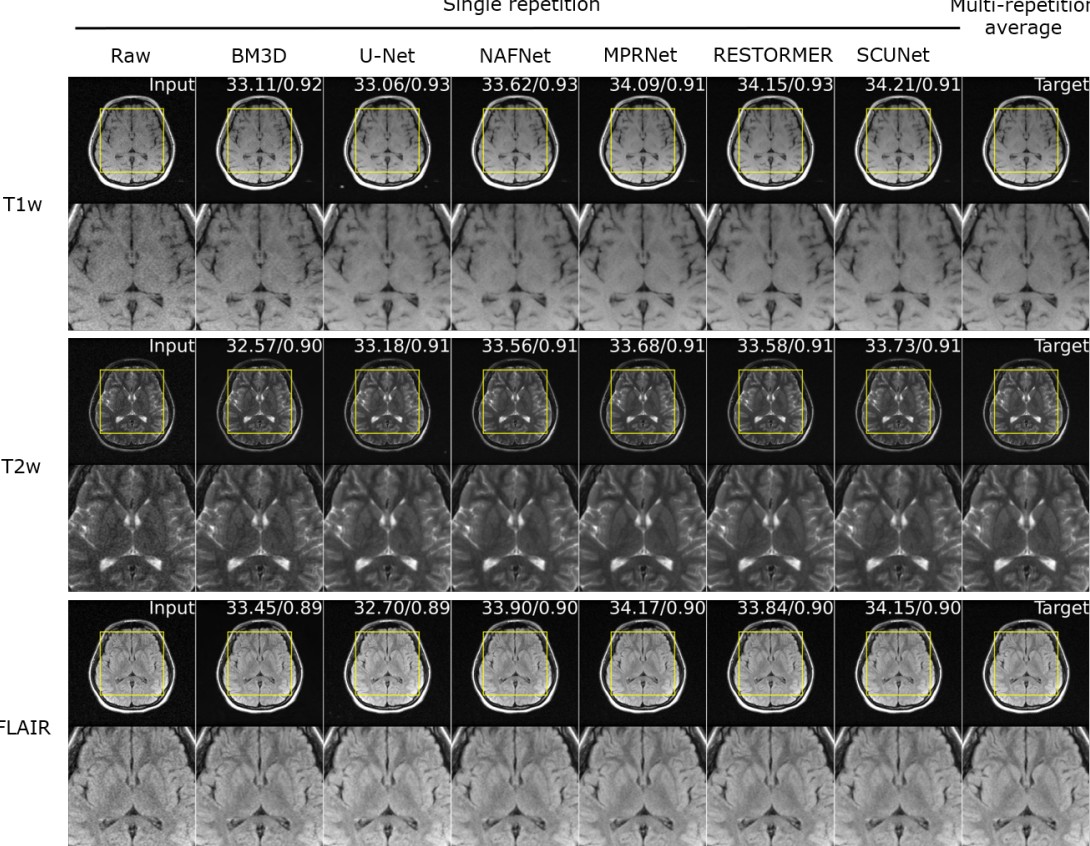

Figure 2: Results of the test subset for various denoising methods. The mean peak signal-to-noise ratio (PSNR) and structural similarity index (SSIM) values were calculated for each contrast on the test subset and labeled on top of the representative images. Most data-driven methods achieved higher PSNR and SSIM values than the traditional BM3D (Wang et al., 2004) method, offering visually improved images with less blurring.

The M4Raw dataset enables development of various data-driven methods for low-field MRI denoising. It can also serve as a benchmark dataset for comparing different methods specific to low-field MRI. Future studies can explore the effect of denoising on downstream applications. Apart from denoising, potential research applications for this dataset include parallel imaging, super-resolution, motion correction, and image style transfer from low-field to high-field.

To facilitate users of this dataset, we have released the following Github repository: https://github.com/mylyu/M4Raw. The repository contains Python examples for data reading and deep learning model training. We welcome all types of queries and suggestions for this dataset.

## Acknowledgments

This work was supported by the National Natural Science Foundation of China (No. 62101348), Shenzhen Higher Education Stable Support Program (No. 20220716111838002), and Natural Science Foundation of Top Talent of Shenzhen Technology University (No. 20200208 and No. GDRC202134).

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
