# OpenReview forum: "Recent Updates of the M4Raw Dataset and Applications in Evaluating MRI Denoising Methods"
_MIDL.io/2024/Short_Papers — MIDL 2024 Short Papers_

### Official Review · Reviewer_tvBx · 2024-04-24

**Confidence:** 5
**Final Rating:** 4

**Review:**

This paper presents new data in the M4Raw dataset, one of the few publicly available datasets out there with complex k-space data. The dataset includes a bunch of repetitions, which makes it an excellent candidate to evaluate image denoising methods (using a single repetition as input, and the average of all repetitions as ground truth).  This is illustrated by a comparison of different existing denoising algorithms using the new data.

Even though there's no new methods / novelty in this paper, I think that the MIDL community will be interested in hearing about the progress made by the authors in developing M4Raw.



Minor comments:

Intro: I wouldn't say that MRI data are scarce; there are boatloads of public datasets with 100s/1000s of scans. I would instead say that MRI datasets with complex / k-space data are scarce.

Data: Please specify resolution of the images.

Figure 1: I would crop tighter around the brain so you can zoom in more and make the differences in noise more visible.

Results: it is always nice to see the effect of denoising on downstream applications. I underestand there's a 3-page limit, but I'd encourage the authors to mention this.

---

### Decision · Program_Chairs · 2024-04-26

Accept